# Deep Transfer Learning for the Multilabel Classification of Chest X-ray Images

**DOI:** 10.3390/diagnostics12061457

**Published:** 2022-06-13

**Authors:** Guan-Hua Huang, Qi-Jia Fu, Ming-Zhang Gu, Nan-Han Lu, Kuo-Ying Liu, Tai-Been Chen

**Affiliations:** 1Institute of Statistics, National Yang Ming Chiao Tung University, Hsinchu 30010, Taiwan; qijia444@gmail.com (Q.-J.F.); eric956412@gmail.com (M.-Z.G.); ctb@isu.edu.tw (T.-B.C.); 2Department of Pharmacy, Tajen University, Pingtung City 90741, Taiwan; ed103911@edah.org.tw; 3Department of Radiology, E-Da Hospital, I-Shou University, Kaohsiung City 82445, Taiwan; 4Department of Medical Imaging and Radiological Science, I-Shou University, Kaohsiung City 82445, Taiwan; 5Department of Radiology, E-Da Cancer Hospital, I-Shou University, Kaohsiung City 82445, Taiwan; ed102500@edah.org.tw

**Keywords:** convolutional neural network, deep learning, source data set, supervised classification

## Abstract

Chest X-ray (CXR) is widely used to diagnose conditions affecting the chest, its contents, and its nearby structures. In this study, we used a private data set containing 1630 CXR images with disease labels; most of the images were disease-free, but the others contained multiple sites of abnormalities. Here, we used deep convolutional neural network (CNN) models to extract feature representations and to identify possible diseases in these images. We also used transfer learning combined with large open-source image data sets to resolve the problems of insufficient training data and optimize the classification model. The effects of different approaches of reusing pretrained weights (model finetuning and layer transfer), source data sets of different sizes and similarity levels to the target data (ImageNet, ChestX-ray, and CheXpert), methods integrating source data sets into transfer learning (initiating, concatenating, and co-training), and backbone CNN models (ResNet50 and DenseNet121) on transfer learning were also assessed. The results demonstrated that transfer learning applied with the model finetuning approach typically afforded better prediction models. When only one source data set was adopted, ChestX-ray performed better than CheXpert; however, after ImageNet initials were attached, CheXpert performed better. ResNet50 performed better in initiating transfer learning, whereas DenseNet121 performed better in concatenating and co-training transfer learning. Transfer learning with multiple source data sets was preferable to that with a source data set. Overall, transfer learning can further enhance prediction capabilities and reduce computing costs for CXR images.

## 1. Introduction

A chest X-ray (CXR), which is generated by exposing the chest to a small dose of ionizing radiation, is a projection radiograph of the chest used for imaging subtle lesions and the density of human tissues. It is commonly used for visualizing the condition of the thoracic cage, chest cavity, lung tissue, mediastinum, and heart. It can thus facilitate the diagnosis of common thorax diseases, including aortic sclerosis or calcification, arterial curvature, abnormal lung fields, anomalous lung patterns, spinal lesions, intercostal pleural thickening, and cardiac hypertrophy.

Computer vision technology and hardware computing capabilities have progressed considerably. Considering the overload of medical resources and the high demand for medical image analysis, the development of computer-aided diagnosis systems with a high diagnostic efficiency and accuracy is warranted. CXR images containing large amounts of physiological data can aid data-hungry deep learning paradigms in the construction of valuable intelligent auxiliary systems. Deep learning is a branch of machine learning; through linear or non-linear transform from multiple layers, deep learning can automatically extract sufficient and representative features from a data set. In traditional machine learning, features are usually extracted using handcrafted rules, which are created by relevant domain experts. After the data characteristics are understood, useful and effective features can be produced. However, the ability to automatically extract features from deep learning can reduce the time spent by experts in feature engineering. Therefore, deep learning may afford an excellent performance in applications where machines may have failed in the past.

Several studies have used deep learning for CXR analysis, particularly for image classification. Most of these studies have trained deep learning models by using well-designed convolutional neural network (CNN) architectures such as VGG [1], GoogleNet [2], ResNet [3], and DenseNet [4]. CNN architecture depth, data augmentation, input preprocessing methods, image size, and pretraining schemes can affect the performance of a deep learning model [5]. No standardized design methodology for improving deep learning model performance has been reported thus far. Most of the relevant studies have focused on comparing the performance of multiple design methodologies for a specific task, rather than reporting novel methodologies [6,7,8]. Some studies have achieved methodological novelty by utilizing methods that can aid in improving model performance. For example, Hu et al. [9] and Wu et al. [10] used an extreme learning machine (ELM) to replace the conventional fully connected layer in a deep CNN for real-time analysis and applied a Chimp optimization algorithm or sine-cosine algorithm to ameliorate the ELM’s ill-conditioning and nonoptimal problems. Wang et al. [11] trained CNN models by using the whale optimization algorithm, which can resolve difficulties related to requiring a considerable amount of manual parameter tuning and parallelizing the training process of traditional gradient descent-based approaches. Khishe et al. [12] proposed an efficient biogeography-based optimization approach for automatically finetuning model hyperparameters (e.g., number of output channels, convolution kernel size, layer type, training algorithm’s learning rate, epoch, and batch size), which are typically selected manually. CXR images are commonly classified as normal or abnormal in the literature [7]. Although CXR can be used to detect multiple diseases of the thorax, few methods have been proposed for classifying multiple disease labels [13].

Applying deep learning methods to CXR image analysis may have promising applications. However, the advancement of automatic image analysis in hindered by several underlying limitations. The main limitation is the lack of large-scale CXR datasets. Although the number of parameters required for deep learning models is considerably large, CXR image training data are limited; this can cause model overfitting. Compared with ordinary images, collecting and labeling CXR images can be difficult and cost intensive. CXR images generated using various instruments with different settings and in different environments cannot be naively analyzed together because this can lead to various errors. Transfer learning [14], which uses the knowledge learned from one task as a starting point for related tasks, can aid in making the best use of different CXR databases. Transfer learning mainly involves using a large amount of open-source data to make up for target data shortages so as to achieve improved performance in model fitting.

In this study, we used one private target data set—1630 chest radiographs provided by the E-Da Hospital, I-Shou University, Taiwan—and three open-source datasets—the ImageNet dataset [15], ChestX-ray dataset [16,17] (with >100,000 chest radiograph images provided by the National Institutes of Health (NIH)), and CheXpert dataset [18] (collected by the Stanford ML Group, comprising nearly 220,000 chest radiographs). The images from the private target data set were labeled by radiology specialists as either being disease-free or containing multiple sites of abnormalities—representing a typical multilabel classification problem. Although ImageNet is sufficiently large for deep learning, most of its 14 million images were dissimilar to those in our private data set; therefore, this data set may not have been able to provide accurate feature representations to classify images in our private data set. ChestX-ray and CheXpert, although modest in size, are more similar to our private data set, and thus, they may be able to support target data training more efficiently.

Transfer learning has been widely applied in CXR deep learning analysis. Most studies have first trained deep CNN models on the large ImageNet dataset for natural image classification, followed by the use of trained weights for initialization to retrain all layers or only retrain the final (fully connected) layer for target CXR image classification [8]. The performance of transfer learning might be affected by several factors such as the sizes of the source and target data set, similarity between these data sets, retraining of all or partial layers in the target task, and CNN architecture [19]. For natural image classification, Azizpour et al. [19] and Cui et al. [20] have reported best practices on how these factors should be set in generic and domain-specific tasks, respectively. However, only a few studies have focused on the factors that affect the transferability of medical image analysis approaches. Tajbakhsh et al. [21] considered four different medical imaging applications and demonstrated that CNNs pretrained on ImageNet performed better and were more robust to the size of the target training data than the CNNs trained from scratch. Gozes and Greenspan [22] pretrained their model on the ChestX-ray data set and demonstrated that the pretrained weights enabled the model to exhibit improved predictions on small-scale CXR data sets compared with the performance of a model pretrained on ImageNet. 

To maximize the performance of transfer learning for CXR image classification, the effects of different transfer learning characteristics in medical image analysis must be systematically investigated. Because of the substantial differences between natural and medical images, we could not apply the knowledge learned for natural images in previous studies [19,20] to our current CXR analysis. Accordingly, our study focused on the multilabel classification of CXR images, which is a crucial topic that warrants research. We thoroughly investigated the aforementioned transferability factors and included the ImageNet, ChestX-ray, and CheXpert data sets as the source data. Previous studies have typically used one type of source data at a time for transfer learning. In this study, we developed new approaches for integrating different source data sets practically to eventually obtain novel powerful source data sets. Our results may aid in devising best practices for the efficient use of different types of data sets to alleviate the insufficiency of training data and enhance the performance of deep learning models in the medical field.

## 2. Materials

The target data set analyzed in this study contained the CXR images from the E-Da Hospital, I-Shou University, Taiwan. A deep learning model was trained to classify these images as being disease-free or containing multiple sites of abnormalities. We selected three source datasets with different sizes to pretrain our deep learning model and to improve its performance for the target data. Of all our source data sets included here, ImageNet contains the largest amount of data, followed by CheXpert and then ChestX-ray. Although ImageNet is the largest in size, its data had less similarity to the target data than the other two data sets had. Table 1 lists the basic characteristics of the data sets used.

### 2.1. Target Data Set

The target data set contained CXR images that were collected from patients who received a CXR between January 2008 and December 2018; the images were stored in the DICOM (Digital Imaging and Communications in Medicine) format. These images were retrospectively extracted from the archiving and communication system (PACS) of E-Da Hospital. Patients’ gender, age, and diagnostic reports from radiology specialists were also provided. Images were excluded if their quality and the corresponding interpretation of diagnostic reports were unclear. Images of minors (aged < 18 years) were also excluded from this study. This clinical study was approved by the Institutional Review Board of E-Da Hospital. All patients signed written informed consent before participating.

The image size ranged between 1824 and 2688 pixels in length and 1536 and 2680 pixels in width. The image resolution was 0.16 mm per pixel. When we experimentally analyzed images resized at 1024×1024, the results were nonsignificant, and the process was cost intensive. Therefore, we resized all images to 512×512 pixels for analysis. 

We first removed duplicate and outlier images. We also discarded five images uniquely labeled as “heart pacemaker placement” due to their inconsistent disease property. The analyzed data set comprised 1630 images with 1 normal label and 17 diseases labels, which had been integrated into 8 categories (including normal) with guidance from physicians. Of these images, 1485 had a single label and 145 had multiple labels. Table 2 lists the numbers of images that contained certain labels in the data set.

### 2.2. Source Data Sets

#### 2.2.1. ImageNet

ImageNet is a large visual database designed for visual object recognition. It contains more than 14 million images that have been hand-labeled to indicate their object category, of which there are >20,000. The data set size is approximately 1 TB. The Python package Keras provides the pretrained weights for various networks, which eliminates the requirement to training them from scratch.

#### 2.2.2. CheXpert

CheXpert is a large CXR image data set collected by the Stanford ML Group; it comprises 224,316 chest radiographs labeled to indicate the presence of 13 common thoracic diseases or labeled “no finding” to indicate the absence of all diseases [18]. Natural language processing (NLP) is used to extract observations from radiology reports, and this extracted information serves as the basis of labeling. The training labels in the data set for each category are 0 (negative), 1 (positive), or u (uncertain). Different approaches for using uncertainty labels during model training may lead to differences in network performance. In this study, we followed the results from the original paper for dealing with the uncertainty label in relation to five diseases: atelectasis, edema, pleural effusion, cardiomegaly, and consolidation. In other words, we reconstructed a five-dimensional label vector for the five aforementioned diseases—where u was replaced with 1 for the first three diseases and with 0 for the final two—and then we applied it as the label for five-class multilabel classification. In other words, we transformed the pretraining task into a task for classifying whether the images had these five diseases. 

In this study, we resized the images to 512×512 pixels to save memory. Moreover, some of the images from the data set were gray-scale images (where they only had one channel), whereas the other images had four channels. To feed them into the available pretrained models through a three-channel input, we replicated the one-channel images three times but removed the fourth channel of the four-channel images.

#### 2.2.3. ChestX-ray

ChestX-ray is an open-source data set compiled by the National Institutes of Health; it comprises 112,120 chest radiographs with 1 normal label and 14 disease labels [16,17]. Similar to CheXpert, ChestX-ray uses NLP for labeling, but it does not have an uncertainty label. We stacked and removed the channels of the images to fit them into the available pretrained networks as we did for CheXpert. We also resized the original images to 512×512 pixels for fast processing and excluded the disease label “hernia” (and thus deleted 110 images) from the analysis because its sample size was relatively small.

## 3. Methods

This analysis aimed to predict common thorax diseases in patients through their CXR images. The images were labeled as being disease-free or containing multiple sites of abnormalities, representing a typical multilabel classification problem. Deep learning architectures were used to automatically extract features and build a classifier, and transfer learning was conducted to combine information from different datasets to improve model performance. We trained deep learning models using the well-designed convolutional neural network (CNN) architectures ResNet50 [3] and DenseNet121 [4]. Implementing transfer learning in the CNN involved reusing the first several layers of the network for classifying images in the open-source data sets ImageNet, CheXpert, and ChestX-ray (the source task). Pretrained weights from these layers served as the starting points for classifying the CXR images from E-Da Hospital (the target task). By combining three source data sets in various manners, we obtained several different sets of pretrained weights. To transfer the pretrained weights to the target task, we either used the pretrained weights as the initial values and retrained the model from the scratch or fixed the weights of some early layers at the corresponding pretrained weights and reconstructed the others.

Image augmentation techniques were applied to artificially create variations in the existing images; this expanded the training data set to represent a comprehensive set of possible images. Our target dataset was imbalanced: the number of images containing each disease label was unequal. Deep learning algorithms can be biased toward the majority labels and fail to detect the minority labels. To prevent imbalanced data, sample weighting in the loss function was used. Model performance was evaluated using various metrics and stratified five-fold cross-validation. Our analytic approach is illustrated in Figure 1. The programming language Python [23] was used to implement these methods.

### 3.1. Multilabel Classification

Image classification aims at building a model that maps the input of the ith image $x_{i}$ to a label vector $y_{i}=\left(y_{i1},\cdots ,y_{iK}\right)$, where $y_{ik}=0\;\mathrm{or}\;1,\;k=1,\cdots ,K$ is the indicator for the kth disease class. Multiclass classification involves classifying an image into one of multiple classes; in other words, the label vector of multiclass classification has only a single element equal to 1. However, in multilabel classification, more than one class may be assigned to the image, with the label vector possibly having 0 or 1 in each element. 

CXR images in our target data set were labeled for seven diseases, and multiple diseases were often identified in one image. The target task was multilabel classification and defined a seven-dimensional label vector with an all-zero vector (0,0,0,0,0,0,0) representing a normal status (where none of the seven diseases was detected). We treated multilabel classification as a multiple binary classification problem; thus, the loss function was the sum of multiple binary cross-entropies. However, our data were seriously imbalanced: the elements of the label vectors were almost equal to 0. The number of 0 s was much larger than that of 1 s; this could have misled our model toward predicting 0 s if the usual loss function was used. Therefore, we adjusted the common binary cross-entropy with weights that considered the proportions of 0 s and 1 s in the same sampling batch. 

Suppose that the entire training set is divided into J batches, with each batch size being M. Let $x_{jm}$ be the input for the mth image in the jth batch and $y_{jm}=\left(y_{jm1},\cdots ,y_{jmK}\right)$ be its label vector. Then, the proposed weighted binary cross-entropy (WBCE) loss is defined as:(1)$$L_{\mathrm{WBCE}}=\overset{J}{\underset{j=1}{\sum }}\left(\overset{M}{\underset{m=1}{\sum }}\left\{\beta_{Pj}\underset{k:\;y_{jmk}=1}{\sum }\left[-\ln\left(\sigma \left(f_{k}\left(x_{jm}\right)\right)\right)\right]+\beta_{Nj}\underset{k:\;y_{jmk}=0}{\sum }\left[-\ln\left(1-\sigma \left(f_{k}\left(x_{jm}\right)\right)\right)\right]\right\}\right),$$
where $f_{k}\left(x_{jm}\right)$ is $x_{jm}$’s kth input for the final fully-connected layer, $\sigma \left(z\right)=e^{z}/\left(1+e^{z}\right)$ is the sigmoid function, and $\beta_{Pj}$ is set to $\frac{\left|P_{j}\left|+\right|N_{j}\right|}{\left|P_{j}\right|}$, whereas $\beta_{Nj}$ is set to $\frac{\left|P_{j}\left|+\right|N_{j}\right|}{\left|N_{j}\right|}$, with $\left|P_{j}\right|$ and $\left|N_{j}\right|$ being the number of 1 s and 0 s in the label vectors of the jth batch, respectively.

### 3.2. Image Augmentation

Training deep CNNs requires a considerable amount of data, but the sample size of medical images is typically not sufficiently large. Image augmentation is a powerful technique for generating images using a combination of affine transformations—such as shift, zoom in-zoom out, rotate, flip, distort, and shade with a hue—which can feed more images into the neural networks and exploit information in the original data more fully. We augmented our training images by using the ImageDataGenerator function in the Keras Python library. “Online” augmentation was applied; here, we applied the image augmentation techniques in mini-batches and then fed them to the model. The model with online augmentation was presented with different images at each epoch; this aided the model in generalizing, and the model did not need to save the augmented images on the disk, which reduced the computing burden.

### 3.3. Deep CNNs

Image classification using deep CNNs has an outstanding performance compared with traditional machine learning approaches. CNN is a deep learning method in which a series of layers is constructed to progressively extract higher-level features from the raw input. In a CNN, the typical layers comprise the input layer (which imports image data for model training), the convolution layer (which uses various filters to automatically learn representation of features at different levels), the pooling layer (which selects the most prominent features to reduce the dimension of subsequent layers), and the fully connected layer (which flattens the matrix feature maps into a single vector for label prediction).

The CNN architecture refers to the overall structure of the network: the types of layers it should have, the number of units each layer type should contain, and the manner in which these units should be connected to each other. CNN architectures such as VGG [1], GoogleNet [2], ResNet [3], and DenseNet [4] have been widely used. We included two high-performing CNN models: 50-layer ResNet (ResNet50) and 121-layer DenseNet (DenseNet121). ResNet50 has fewer filters and a lower complexity than VGG nets, and DenseNet121 requires fewer parameters than ResNet50 does.

#### 3.3.1. ResNet

In theory, the deeper the network model, the better its results. Nevertheless, the degradation problem may arise: as the network becomes deeper, the model’s accuracy may become saturated or even decrease. This problem is different from overfitting because of increased training errors. ResNet made a historical breakthrough in deep neural networks by solving the degradation problem through residual learning. 

In this study, we established a deeper model by stacking new layers on a shallower architecture. Let x denote the output of the shallow part of the model and H(x) denote the output of the deeper model. No higher training error should be obtained if the added layers are for identity mapping. Rather than expecting the stack of layers to learn identity mapping, ResNet argues that it is easier to let these layers fit a residual mapping of F(x)=H(x)−x to zero and recast the output of the deeper model as $\hat{F}\left(x\right)+x$, where $\hat{F}\left(x\right)$ is the fitted residual. Generally, $\hat{F}\left(x\right)$ will not be zero; therefore, the stacking layers can still learn new features and demonstrate an improved performance.

The formulation of $\hat{F}\left(x\right)+x$ can be realized using feedforward neural networks with a shortcut connection, which skips some of the layers in the neural network and feeds the output of one layer as the input to the next layers. A series of shortcut connections (residual blocks) forms ResNet. This study applied the deeper ResNet50 that contained 50 layers and used a stack of three layers with 1×1, 3×3, and 1×1 convolutions as the building residual block. This three-layer residual block adopted a bottleneck design to improve computational efficiency, where the 1×1 layers were responsible for reducing and then increasing (restoring) the dimensions, leaving the 3×3 layer as a bottleneck with smaller input and output dimensions [3]. In ResNet50, batch normalization (BN) [24] is adopted immediately after each convolution and before ReLU activation, and global average pooling (GAP) [25] is performed to form the final fully connected layer.

#### 3.3.2. DenseNet

As CNNs become increasingly deep, information about the input passes through many layers and can vanish by the time it reaches the end of the network. Different approaches, which vary in network topology and training procedure, create short paths from early layers to later layers to address this problem. DenseNet proposes an architecture that distills this insight into a simple connectivity pattern of dense blocks and transition layers. A dense block is a module containing many layers connected densely with feature maps of the same size. In a dense block, each layer obtains additional inputs from all preceding layers, and it passes on its own feature maps to all the subsequent layers. The transition layer connects two adjacent dense blocks, and it reduces the size of the feature map through pooling. Compared with ResNet that connects layers through element-level addition, layers in DenseNet are connected by concatenating them at the channel level.

In a dense block, the convolution for each layer produces k feature maps, which denote k channels in the output. k is a hyperparameter known as the growth rate of DenseNet, which is usually set to be small. Under the assumption that the initial number of channels is $k_{0}$, the number of input channels in the ℓth layer is $k_{0}+k\left(\ell -1\right)$. As the number of layers ℓ increases, the input can be extremely large even if k is small. Because the input of the latter layer grows quickly, the bottleneck design is introduced into the dense block to reduce the burden of calculation, where a 1×1 convolution and then a 3×3 convolution are applied to each layer to generate the output. Similar to ResNet, DenseNet uses a composite of three consecutive operations for each convolution: BN+ReLU+convolution.

#### 3.3.3. Implementation

We used the Python package Keras to implement ResNet50 and DenseNet121. Optimizer Adam [26] with a mini-batch size of 16 and an epoch number of 30 was used. The learning rate started from 0.0001 and was divided by 10 when the validation loss did not decrease in 10 epochs. DenseNet121 set the growth rate to k=12.

### 3.4. Transfer Learning

In practice, we usually do not have sufficient data to train a deep and complex network such as ResNet or DenseNet. Techniques such as image argumentation are insufficient for resolving this problem. The lack of data can cause overfitting, which can make our trained model overly rely on a particular data set and then fail to fit to additional data. This problem can be resolved using transfer learning [14]. The main concepts of transfer learning are to first train the network with sufficiently large source data and then transfer the structure and weights from this trained network to predict the target data. 

In the current study, we used the data from the ImageNet [15], ChestX-ray [16,17], and CheXpert [18] data sets as the source data and the CXR images from the E-Da Hospital as the target data. ImageNet contains more than 14 million natural images and thus is sufficiently large for a deep learning application; however, it does not contain images with similarity to medical images, and therefore, it may fail to provide useful feature representations to classify our target data. ChestX-ray and CheXpert—consisting of about 100,000 and 220,000 CXR images, respectively—are modest in size but are more similar to our target data set.

Implementing transfer learning in the CNN involves reusing the parameter estimators pretrained on the source data when fitting the target data. Model finetuning is a method in which these pretrained parameter estimators are used as the initial values and finetuned to fit the target data. In the layer transfer approach, target data fitting preserves some of the layers from a pretrained model and reconstructs the others. We thus adopted model finetuning or layer transfer to reuse the parameter estimators pretrained on the source data (from ImageNet, ChestX-ray, or CheXpert).

We next tried to combine these source data sets to obtain new powerful source data sets for transfer learning. 

#### 3.4.1. Initiating Transfer Learning

In Keras, we can implement transfer learning with initial weights selected randomly or from ImageNet pretraining. ImageNet pretraining may result in robust parameter estimation due to the diversity and richness of the data set, and the pretraining of the similar data sets ChestX-ray and CheXpert may accelerate the convergence of the model. Therefore, we adopted five different pretrained weights in modeling our target data: ImageNet pretraining with randomly selected initials, ChestX-ray pretraining with random or ImageNet pretraining initials, and CheXpert pretraining with random or ImageNet pretraining initials.

#### 3.4.2. Concatenating Transfer Learning

In this approach, we first constructed two ResNet backbone models by using different pretrained weights: one from ImageNet pretraining with random initials (IR) and the other from ChestX-ray pretraining with random initials (CR). Thereafter, we concatenated the outputs of the final convolution layer from two models and then used the GAP to reduce the dimension, and finally applied the fully connected layer to generate the final prediction (Figure 2). The ResNet model concatenating IR and CheXpert pretraining with random initials (XR) was obtained in a similar manner. The DenseNet backbone models concatenating IR and CR and concatenating IR and XR could also be obtained. Models using different pretrained weights may extract different features. In contrast to the first approach aimed at extracting features from a single domain, this approach could expand features from two different domains. However, compared with the first approach, this approach used twice the memory and time to store and upgrade parameters.

#### 3.4.3. Co-Training Transfer Learning

In this approach, we aimed to combine two source data sets, namely ChestX-ray and CheXpert, to a larger CXR data set to serve as source data for medical image classification tasks. ChestX-ray and CheXpert cannot be combined directly: although both the data sets contain only CXR images, they use different class definitions. To resolve this issue, we applied a co-training approach. First, we fed images from two source data sets to jointly train convolutional layers but to also connect to different fully connected layers to predict their corresponding classes. In other words, they shared the parameters from the convolution layers but not those after the convolution layers. Finally, we reserved these shared convolutional weights as the pretrained weights in target data model fitting. The reason that we did not cotrain ImageNet with ChestX-ray (or CheXpert) was that merging different domain data may mislead the model and reduce its efficiency. The co-training approach is illustrated in Figure 3.

### 3.5. Evaluation

#### 3.5.1. Stratified K-Fold Cross-Validation

In K-fold cross-validation, first, the data are shuffled to ensure the randomness when the data are split. Second, whole data are split into K groups, where K is usually set to be 5 or 10 based on the sample size. Third, one of groups is considered “test data,” and the others are considered “training data” with a total of K different test-training combinations (K cross-validation rounds). Finally, the “training data” are further divided into “training” and “validation” sets, which are used to train the model parameters and to instantly evaluate the performance of various hyper-parameters, respectively.

In stratified K-fold cross-validation, every group must have the same class distribution. This method has great applicability for multiclass classification, where each sample belongs to only one class. In a multilabel task, every sample may have multiple class categories. To appropriately perform stratified K-fold cross-validation for a multilabel classification task, performing iterative stratification is essential [27]; it can be implemented using Python’s scikit-learn-compatible package MultilabelStratifiedKFold.

#### 3.5.2. Metrics

For an imbalanced data set, good accuracy does not indicate that a classifier is reliable. A receiver operating characteristic (ROC) curve is a graphical plot that illustrates the performance of a binary classifier (with vs. without the label) at different thresholds, where the x-axis represents the false positive rate, and the y-axis represents the true positive rate. The area under the ROC curve (AUC) is used to summarize the ROC curve. A precision (PR) curve is another type of plot that evaluates the performance of a model, with the x-axis being the recall and the y-axis being the precision. The area under the PR curve is the average precision (AP). Moreover, training accuracy represents the average accuracy of the training sets during the five cross-validation rounds. Test accuracy, AUC, and AP refer to the accuracy, AUC, and AP of the five test groups, respectively. In multilabel classification, we can treat the problem as a combination of multiple binary classifications; thus, we can evaluate every label individually with binary metrics and obtain the average of these binary metrics (i.e., the mean metric). In this study, we calculated the training accuracy, test accuracy, test AUC, and test AP for each individual label and the mean training accuracy, test accuracy, test AUC, and test AP for all labels.

## 4. Results

This section presents the results of all the experiments performed in the current study. Table 3 presents a summary of our experimental configurations and parameter settings.

### 4.1. Layer Transfer versus Model Finetuning

To using the layer transfer approach, we fixed some lower layers in the network at weights from the source data and retrained the remaining layers on the target data. We first froze 10, 22, 40, and 49 layers in ResNet50, which corresponded to the end of the first, second, third, and fourth residual blocks. We used weights pretrained on ImageNet with random initials, ChestX-ray with random initials, and ChestX-ray with ImageNet pretraining initials. 

The mean test accuracies and AUCs from the layer transfer or model finetuning on different pretrained weights are listed in Table 4. When adopting the layer transfer approach, it is not recommanded to modify only the final layer, which is equivalent to freezing the first 49 layers in ResNet50. Here, for pretrained weights from ChestX-ray with random or ImageNet pretraining initials, freezing more layers typically led to a significantly higher accuracy but a somewhat worse AUC. For the ImageNet pretrained weight, freezing more layers resulted in a lower accuracy and AUC, although the difference was negligible. These results indicated that the ideal number of frozen layers depends on the similarity between the source and target data. When the target data are distinct from the source data, more layers may need to be unfrozen.

Compared with the layer transfer approach, model finetuning afforded a higher accuracy when the ImageNet pretrained weight was set directly and provided considerably larger AUCs for all types of pretrained weights. The model finetuning approach appeared to an attractive option for focusing on prediction accuracy related to every disease label without to sacrificing the minority label.

### 4.2. Effects of Transfer Learning

After transfer learning was applied with model finetuning and pretrained weights from ImageNet, ResNet50′s mean test accuracy increased by 11% and the mean test AUC increased by 4% compared with when transfer learning was not applied (Table 4). After this transfer learning approach was used to train DenseNet121, the mean test accuracy and mean test AUC were 0.799 and 0.803, respectively. However they decreased to 0.657 and 0.736, respectively, after DenseNet121 was trained without transfer learning. With transfer learning, DensNet121’s accuracy increased by 18% and the AUC increased by 8%. Model finetuning-based transfer learning could improve the model fit and appeared to have a greater influence on models with a more complex structure (i.e., those with a greater network depth). These performance improvements remained when model finetuning with pretrained weights from ChestX-ray (with random or ImageNet pretraining initials) was applied.

The layer transfer approach did not always improve model performance (Table 4). Even with the best selected number of frozen layers, models with transfer learning had worse AUCs than those without transfer learning.

### 4.3. Comparison of Various Transfer Learning Approaches

The results from Section 4.1 and Section 4.2 demonstrated that compared with layer transfer, model finetuning—re-training the whole model with pretrained weights as initial values—provided a better prediction model. We thus adopted model finetuning for subsequent analyses. 

This section presents the results from initiating, concatenating, and co-training transfer learning methods, which use different approaches to combine several source data sets to provide an improved model performance. This performance, evaluated using mean accuracy, AUC, and AP on training and test data, is presented in Table 5. In the subsequent subsections, we compare this performance from the perspective of backbone models, source data sets, and combined methods.

#### 4.3.1. Backbone Model Comparison

In this study, we used ResNet50 and DenseNet121 as the backbone model for transfer learning. Compared with DenseNet121, ResNet50 required less time to train a model, but it had more parameters to save. To summarize the two backbone models’ performance levels, the radar plots for mean test AUCs and APs from various transfer learning approaches were created (Appendix A). In initiating transfer learning, ResNet50 outperformed DenseNet121. By contrast, DenseNet121 performed better in concatenating and co-training transfer learning. Therefore, a model with a relatively complex structure may provide more accurate results if a larger, more diverse source data set created by combining different data sets is applied. 

#### 4.3.2. Source Data Comparison

In this study, we used data from three sources—ImageNet, ChestX-ray, and CheXpert—to perform transfer learning. Each has its own strengths and limitations. Although it is nearly 100 times the size of other two data sets, ImageNet’s data had the weakest association with our target data. Although CheXpert is twice the size of ChestX-ray, it contains uncertainty labels, which might increase the difficulty of the training process. ChestX-ray is the smallest data set among those included in this study; nevertheless, it demonstrated the stronger connection to our target data and the most precise labeling. To compare the performance of the different source data, we collected the results from ResNet50 with initial weights from IR, CR, and XR. Their PR and ROC curves for each disease label on test data are presented in Appendix A. DenseNet121’s corresponding PR and ROC curves are presented in Appendix A.

For both the backbone models, the use of ChestX-ray as source data led to a better performance than using CheXpert (Table 5). Even though ChestX-ray is only half the size of CheXpert, its accurate labeling made up for the lack of data. Although ImageNet had the best performance in the training process, its performance in the test process was worse than that of ChestX-ray when it was fit in the ResNet50 model (Table 5). Therefore, the use of ImageNet as the source data possibly led to overfitting. In general, the ChestX-ray data had the best performance when only one single-source data set was adopted as in previous studies. 

#### 4.3.3. Comparison of Combined Methods

We used three methods to combine source data sets. The initiating transfer learning approach adopted ImageNet initials when training pretrained weights from ChestX-ray or CheXpert data to ensure a robust estimation of these pretrained weights. Concatenating transfer learning aimed at collecting the features obtained using distinct source data sets (e.g., ImageNet and ChestX-ray) to expand the features’ coverage. Finally, co-training transfer learning combined two similar source data sets (e.g., ChestX-ray and CheXpert) to train pretrained weights, with the assumption that a large data size can improve model performance.

The combining methods’ PR and ROC curves for an individual disease label on test data are shown in Appendix A. In summary, first, for both backbone models, using ChestX-ray with random initials as the source data led to a better performance than using ChestX-ray with ImageNet initials, and using CheXpert with ImageNet initials as source data led to a better performance than using CheXpert with random initials. ImageNet initials ensured a robust estimation of pretrained weights in CheXpert but not in ChestX-ray. Second, concatenating transfer learning provided the highest training accuracy, but it did not achieve the best performance in the test process, possibly because of the overloading of the model with too many parameters (i.e., with twice the number of parameters), causing overfitting. Third, initiating transfer learning was the most suitable for ResNet50, whereas concatenating and co-training transfer learning were the most suitable for DenseNet121. Notably, under-fitting may have arisen for co-training transfer learning in DenseNet121: more epochs in training can overcome this issue. Fourth, DenseNet121 that involved transfer learning with various source data sets performed better than that with a single-source data set. ResNet50 provided similar results; however, the effect was not as significant as that in DenseNet121. 

## 5. Discussion

When reusing pretrained weights in transfer learning, the approach that re-trains the whole model with pretrained weights as initial values (i.e., the model finetuning approach) typically afforded excellent results but required many more computational resources compared with the other approaches. The layer transfer approach, which freezes some layers on pretrained weights, demonstrated advantages over the model finetuning approach by allowing larger batch sizes and requiring a shorter run time and less GPU/CPU memory. For building the most cost-effective model by using the layer transfer approach, the number of frozen layers should be determined accurately. Our results demonstrated that the higher the similarity between the source and target data, the larger the allowed number of frozen layers should be. However, none of the included approaches were universally applicable; selecting the most appropriate approach would require the assessment of its benefits and costs on the basis of specific goals and available resources.

When adopting only one single-source dataset, ChestX-ray demonstrated a better performance than CheXpert. ChestX-ray is only half the size of CheXpert; nonetheless, its accurate labeling was found to make up for its smaller size. However, after ImageNet initials were attached, CheXpert outperformed ChestX-ray. As such, ImageNet appeared to enhance the data volume and variety of these data sets. 

ResNet50 was suitable for initiating transfer learning, whereas DenseNet121 performed better in concatenating and co-training transfer learning. Transfer learning combined with various source data sets was also preferable with the use of a single-source data set; however, DenseNet121 led to greater benefits than ResNet50 did. Compound weights from several source data sets may be superior to single weights because they contain additional information offered by another data set. Nevertheless, a more complex transfer process may produce more noise. Compared with ResNet50, DenseNet121 was more complex, and its dense block mechanism could process more data and absorb more information; consequently, DenseNet121 is more suitable than BesNet50 for integrating source data sets.

Few studies have focused on the factors that affect the performance of transfer learning in medical image analysis. The strength of the present study lies in its systematic approach to investigating the transferability of CXR image analytic approaches. Nevertheless, this study had several limitations that should be resolved in future studies. First, our experiments were based on the 50-layer ResNet and 121-layer DenseNet architectures, and the derived weights were estimated using the gradient descent-based optimizer Adam [26] under the consideration of manually selected hyperparameter values. ResNet and DenseNet are both widely used for conducting deep learning analyses on CXR images; nevertheless, they differ in several aspects, and such differences can be leveraged to investigate the impact of the CNN architecture on transfer learning. Alternatively, new architectures—such as EfficientNet [28] and CoAtNet [29], which have shown a high performance in challenging computer vision tasks—could be used for analysis. Moreover, to enhance the efficiency of model parameter estimation, Adam may be replaced with recent optimizers such as Chimp [9] and Whale [11], and biogeography-based optimization can be applied to automatically finetune model hyperparameters [12]. Second, CXR images can be taken in posteroanterior, anteroposterior, and lateral views. In this study, we included only one target data set in which all CXR images were in the posteroanterior view; this may have limited the real-world applicability of our findings. Moreover, we included only image classification as the target task. However, CXR images can be used for several other types of deep learning tasks such as segmentation, localization, and image generation [5]. Accordingly, future studies could use CXR images taken in different positions and could consider a wide range of deep learning tasks. 

## 6. Conclusions

In this study, we conducted a thorough investigation of the effects of various transfer learning approaches on deep CNN models for the multilabel classification of CXR images. Our target data set, collected through general clinical pipelines, contained 1630 chest radiographs with 17 clinically common disease labels. Transfer learning methods that reused pretrained weights through model finetuning and layer transfer were examined. We considered three source data sets with different sizes and different levels of similarity to our target data and assessed their effect on transfer learning effectiveness. These source data sets could be incorporated into transfer learning individually or in combination. We also proposed initiating, concatenating, or co-training different source data sets for joint transfer learning and used two backbone CNN models with different network architectures to adopt the aforementioned transfer learning approaches. 

Several substantial findings were obtained. The results demonstrated that transfer learning could improve the model fit. Compared with the layer transfer approach, the model finetuning approach typically afforded better prediction models. When only one single-source data set was adopted as in previous studies, ChestX-ray outperformed ImageNet and CheXpert. However, CheXpert with ImageNet initials attached performed better than ChestX-ray with ImageNet initials attached. ResNet50 performed better in initiating transfer learning, whereas DenseNet121 performed better in concatenating and co-training transfer learning. Transfer learning with multiple source data sets was preferable to that with a single-source data set.

## Figures and Tables

**Figure 1 diagnostics-12-01457-f001:**
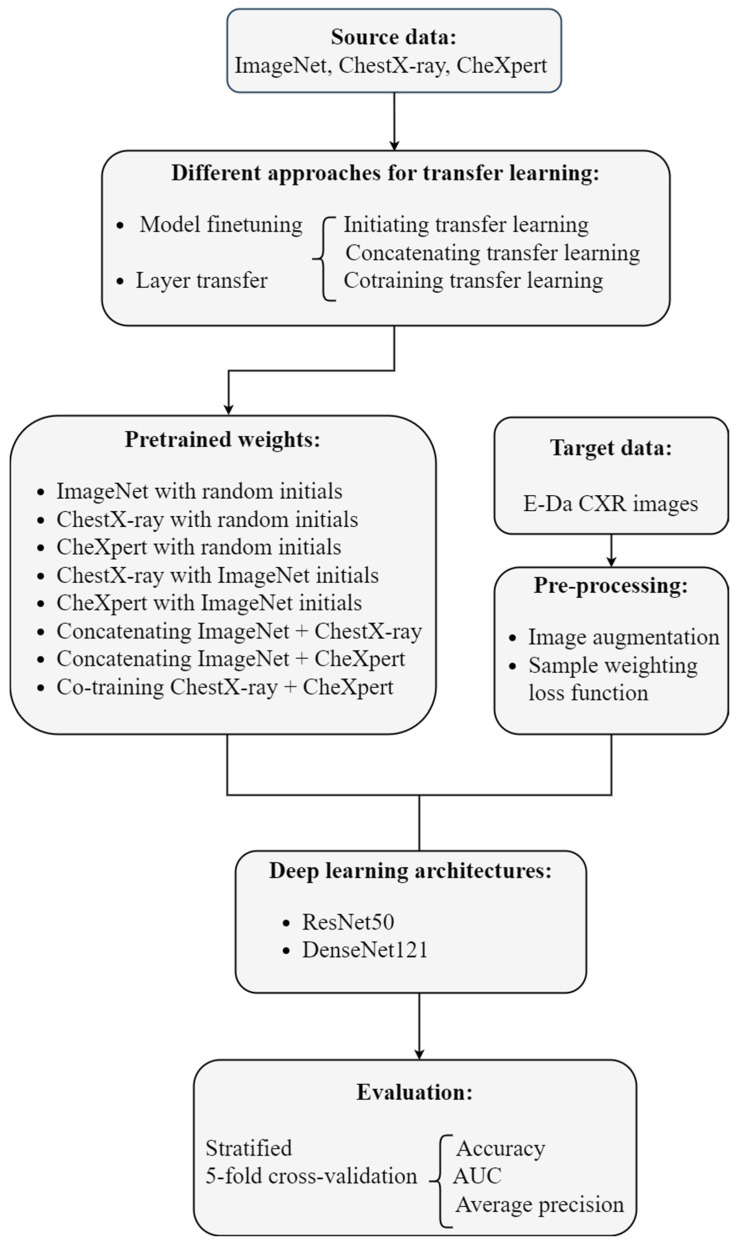
Flow chart of our deep transfer learning approach for the multilabel classification of the chest X-ray images.

**Figure 2 diagnostics-12-01457-f002:**
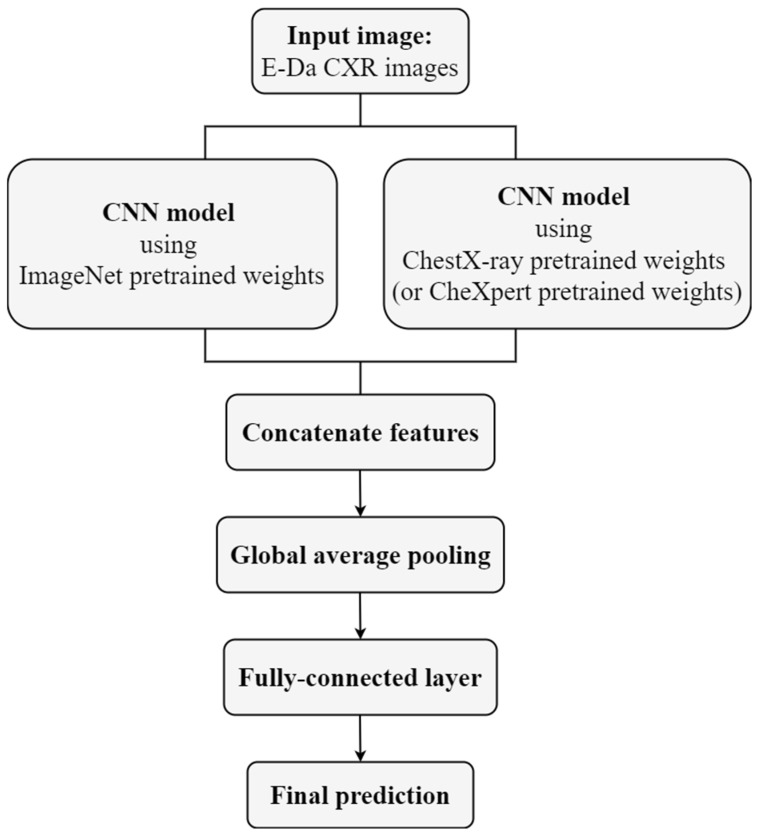
Flow chart of concatenating transfer learning.

**Figure 3 diagnostics-12-01457-f003:**
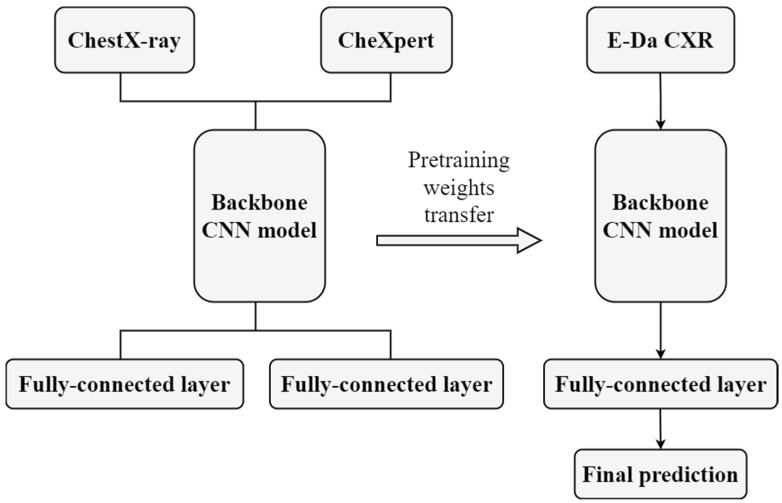
Flow chart of co-training transfer learning.

**Table 1 diagnostics-12-01457-t001:** Characteristics of the included data sets.

Category	Name	Label Category	Size	Feature
Source data	ImageNet	20,000 +	14 million +	Large and diverse
Source data	CheXpert	14 ^1^	224,316	Similar to the target data
Source data	ChestX-ray	14 ^2^	112,010 ^4^	Similar to the target data
Target data	E-Da chest X-ray	8 ^3^	1630	Small but important

^1^ Thirteen common thoracic disease labels and a “no finding” label (indicating the absence of any disease). ^2^ Thirteen disease labels and a normal label. ^3^ Seven disease labels and a normal label. ^4^ One hundred and ten images labeled as “hernia” in the original dataset were discarded in this analysis due to the small sample size.

**Table 2 diagnostics-12-01457-t002:** Numbers and labels of images in the target data set.

Category	Sample Size	Subcategory	Sample Size
normal	1212	normal	1212
aortic sclerosis/calcification	90	aortic arch atherosclerotic plaque	90
arterial curvature	93	tortuous aorta/thoracic aortic ectasia	93
abnormal lung fields	33	shadows of pulmonary nodules	13
		consolidations or lung cavities in the upper lobes	5
		pulmonary fibrosis	15
increased lung patterns	153	atelectasis/focal consolidation	109
		enlarged hilar shadow	44
spinal lesions	148	degenerative joint disease of the thoracic spine	75
		scoliosis	73
cardiomegaly	41	cardiomegaly	41
intercostal pleural thickening	36	intercostal pleural thickening	36

**Table 3 diagnostics-12-01457-t003:** Summary of experimental configurations and parameter settings.

Configuration	Setting
Target data set	CXR images from the E-Da Hospital, Taiwan
Target task	Multilabel classification of CXR images
Loss function	Weighted binary cross entropy
Image augmentation	The ImageDataGenerator function in the Keras Python library for online augmentation
Deep CNN modeling	
Backbone architecture	The Python package Keras to implement 50-layer ResNet and 121-layer DenseNet
Optimizer	Adam
Mini-batch size	16
Epoch number	30
Learning rate	Start from 0.0001 and is divided by 10 when the validation loss does not decrease in 10 epochs
Growth rate in DenseNet	12
Transfer learning	
Source data set	ImageNet, ChestX-ray, and CheXpert
Reuse pretrained weights	Model finetuning and layer transfer
Combine source data sets	Initiating, concatenating, and co-training
Evaluation	
Data splitting	The Python package MultilabelStratifiedKFold for stratified K-fold cross-validation
Metrics	Accuracy, AUC, and AP

**Table 4 diagnostics-12-01457-t004:** Mean test accuracies and AUCs of different approaches of reusing pretrained weights in ResNet50 ^1^.

	Pretrained Weight							
	ImageNet with Random Initials		ChestX-ray with Random Initials		ChestX-ray with ImageNet Initials		Random Initials ^2^	
Method	AC ^4^	AUC	AC	AUC	AC	AUC	AC	AUC
Layer transfer ^3^							0.775	0.765
RB_1	**0.841**	0.487	0.557	**0.519**	0.461	**0.512**		
RB_2	0.828	0.488	0.748	0.514	0.749	0.511		
RB_3	0.814	0.469	**0.871**	0.496	**0.851**	0.505		
RB_4	0.771	**0.520**	0.512	0.498	0.646	0.499		
Model finetuning	**0.873**	**0.796**	**0.811**	**0.831**	**0.789**	**0.827**		

^1^ Bold numbers indicate the top two approaches in each metric. ^2^ The model without transfer learning. ^3^ RB_1 = Pretrained weights on the first 10 layers, which correspond to the end of the first residual block; RB_2 = Pretrained weights on the first 22 layers, which correspond to the end of the second residual block; RB_3 = Pretrained weights on the first 40 layers, which correspond to the end of the third residual block; RB_4 = Pretrained weights on the first 49 layers, which correspond to the end of the fourth residual block. ^4^ AC = Accuracy.

**Table 5 diagnostics-12-01457-t005:** Training and test performance of various transfer learning approaches ^1^.

Backbone Model	Transfer Learning Approach ^2^	Mean Training Accuracy	Mean Test AUC	Mean Test AP
Initiating transfer learning				
ResNet50	IR	**0.935**	0.796	**0.214**
ResNet50	CR	0.879	**0.831**	0.209
ResNet50	CI	0.868	**0.827**	0.206
ResNet50	XR	0.819	0.806	0.191
ResNet50	XI	0.852	**0.831**	**0.213**
DenseNet121	IR	0.916	0.803	0.204
DenseNet121	CR	0.807	0.800	0.171
DenseNet121	CI	0.784	0.779	0.179
DenseNet121	XR	0.799	0.781	0.169
DenseNet121	XI	0.864	0.826	**0.221**
**Concatenating transfer learning**				
ResNet50	I+C	**0.935**	0.780	0.207
ResNet50	I+X	0.930	0.776	0.190
DenseNet121	I+C	**0.935**	0.802	0.210
DenseNet121	I+X	0.914	0.813	0.210
**Co-training transfer learning**				
ResNet50	C∪X	0.855	0.790	0.191
DenseNet121	C∪X	0.775	0.826	0.210

^1^ Bold numbers indicate the top three approaches in each metric. ^2^ IR = ImageNet pretraining with random initials, CR = ChestX-ray pretraining with random initials, CI = ChestX-ray pretraining with ImageNet pretraining initials, XR = CheXpert pretraining with random initials, XI = CheXpert pretraining with ImageNet pretraining initials, I+C = Concatenating ImageNet + ChestX-ray, I+X = Concatenating ImageNet + CheXpert, C∪X = Co-training ChestX-ray + CheXpert.

## Data Availability

The data used and analyzed in this study are available from the corresponding author upon reasonable request.

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
