# Peer review of "Deep Transfer Learning for the Multilabel Classification of Chest X-ray Images"

_diagnostics, 2022, doi:10.3390/diagnostics12061457_

Round 1

Reviewer 1 Report

the paper can be accepted in the current form

Author Response

Manuscript ID: diagnostics-1764304
Title: Deep Transfer Learning for Multilabel Classification of Chest Xray Images
Responses to the Reviewer’s Comments
Introduction
We thank reviewers for their valuable comments, which point out important areas of improvement for the manuscript. We now provide a point-by-point response to the reviewer’s comments. We highlight the changes within the manuscript by using red text.

Response to Reviewer 1 Comments

Point 1: the paper can be accepted in the current form

Response 1: We are pleased to hear that the reviewer recommends the manuscript to be accepted.

Reviewer 2 Report

Many thanks for the opportunity to Review again the manuscript entitled: Deep Transfer Learning for Multilabel Classification of Chest X-ray Images. I have noted some improvements. However, some other major improvements are required. The items that authors have considered in the pathologic chest x ray are not all correct also in the chest X ray nomenclature. Therefore, authors should edit some of them.

Thoracic spine scoliosis is not a finding of arterial curvature. It should be deleted. In abnormal fields: pulmonary nodules are reported twice. Therefore, consider only shadow of pulmonary nodules.

In the increased lung pattern authors should specify what they intend for lung field infiltration because it is not a term recommended to use from the Fleishener Society. I suggest to edit it. Authors should also specify what they intend also for obvious hilar region and increased lung streak. I suggest to edit them. I suggest to consider pulmonary fibrosis in the increased lung pattern and to not use the term of cardiac hypertrophy on a chest x ray but cardiomegaly.

Author Response

Manuscript ID: diagnostics-1764304
Title: Deep Transfer Learning for Multilabel Classification of Chest Xray Images
Responses to the Reviewer’s Comments
Introduction
We thank reviewers for their valuable comments, which point out important areas of improvement for the manuscript. We now provide a point-by-point response to the reviewer’s comments. We highlight the changes within the manuscript by using red text.

Response to Reviewer 2 Comments
I have noted some improvements. However, some other major improvements are required. The items that authors have considered in the pathologic chest x ray are not all correct also in the chest X ray nomenclature. Therefore, authors should edit some of them.

Point 1: Thoracic spine scoliosis is not a finding of arterial curvature. It should be deleted.

Response 1: We are sorry for the error. This subcategory should be “thoracic vertebral artery curvature”.

Point 2: In abnormal fields: pulmonary nodules are reported twice. Therefore,
consider only shadow of pulmonary nodules.

Response 2: Two subcategories “small pulmonary nodules” and “shadows of
2 pulmonary nodules” in the original table have been merged into a subcategory “shadows of pulmonary nodules”.

Point 3: In the increased lung pattern authors should specify what they intend for lung field infiltration because it is not a term recommended to use from the Fleishener Society. I suggest to edit it. Authors should also specify what they intend also for obvious hilar region and increased lung streak. I suggest to edit them.

Response 3: We have combined/modified original subcategories “increased lung
streak” and “lung field infiltration” to be subcategory “increased lung opacity
lesions”. Also, original subcategory “obvious hilar region” is renamed as “enlarged hilar shadow”.

Point 4: I suggest to consider pulmonary fibrosis in the increased lung pattern and to not use the term of cardiac hypertrophy on a chest x ray but cardiomegaly.

Response 4: If considering pulmonary fibrosis in the increased lung pattern category, we change the chest X-ray images that make up the disease labels of our target data set. As a result, we will have to redo all the analyses in the paper. The radiologists from our research group consider current categorization to be reasonable based on the clinical practice in Taiwan. We thus decide not to move “pulmonary fibrosis” from the “abnormal lung fields” category to the “increased lung patterns” category. However, we also add a note to Table 2 to remind readers our categorization. We have renamed “cardiac hypertrophy” as “cardiomegaly”. We thus modify the legends for supplementary figures S3-S18 accordingly

Round 2

Reviewer 2 Report

Many thanks for some adjustments. I suggest to delete the sentence This study had considered pulmonary fibrosis in the abnormal lung fields category although some research suggests in the increased lung pattern and to remain the pulmonary fibrosis in the abnormal lung field because as authors have explained it in the response letter. I suggest to specify also the lung opacity lesions.  However the toracic vertebral curvature I think that it referred to the thoracic aorta is a normal finding and I suggest to not consider and include it.

Author Response

Manuscript ID: diagnostics-1764304
Title: Deep Transfer Learning for Multilabel Classification of Chest Xray Images
Responses to the Reviewer’s Comments

Introduction
We thank reviewers for their valuable comments, which point out important areas of improvement for the manuscript. We now provide a point-by-point response to the reviewer’s comments. We highlight the changes within the manuscript by using red text.

Response to Reviewer 2 Comments

Point 1: I suggest to delete the sentence This study had considered pulmonary fibrosis in the abnormal lung fields category although some research suggests in the increased lung pattern and to remain the pulmonary fibrosis in the abnormal lung field because as authors have explained it in the response letter.

Response 1: Thanks for the suggestion. The sentence “This study had considered
pulmonary fibrosis …” in the notes of Table 2 has been removed in the revised
manuscript.

Point 2: I suggest to specify also the lung opacity lesions.

Response 2: “increased lung opacity lesions” is replaced by “atelectasis/focal
consolidation” in the revised Table 2.

Point 3: However the thoracic vertebral curvature I think that it referred to the thoracic aorta is a normal finding and I suggest to not consider and include it.

Response 3: Thanks for the reminding. We go through our original diagnostic
documents and find some errors due to our negligence. Our original diagnoses were written in Chinese. We then translated them from Chinese to English. In Chinese diagnosis, we used “aorta” or “thoracic aorta” interchangeably, which caused confusion in translation and created an extra subcategory “thoracic vertebral artery curvature”. We now combine two subcategories “tortuous aorta/thoracic aortic ectasia” and “thoracic vertebral artery curvature” into one, which is renamed as “tortuous aorta/thoracic aortic ectasia.”

Round 3

Reviewer 2 Report

Many thanks for the adjustments

This manuscript is a resubmission of an earlier submission. The following is a list of the peer review reports and author responses from that submission.

Round 1

Reviewer 1 Report

This paper proposes Deep Transfer Learning on Multi-Label Classification of Chest X-ray Images. Although the paper has some merits, it still suffers from the issues of 1) insufficient representation and 2) limited experiments. Some comments that might help the authors further improve the quality are summarized as follows:

  • The authors need to go through the entire manuscript to double-check accuracy/consistency of each equation, table, figure, and reference, and ensure English grammar errors-free.
  • In terms of quality of communication, a linguistic expert needs to review the paper and eliminate all grammatical errors. There are quite a lot of typos
  • I have difficulty seeing any novelty in this paper to publish as a new research manuscript in the journal. Actually, considering the fact the Transfer Learning on Multi-Label Classification was introduced by previous research work, what is the novelty of this paper?
  • The literature review needs to be extended. The following paper should be considered  to give an opportunity to readers about recent real-life applications of recent deep learning approach in X-Ray images: Real‑time COVID-19 diagnosis from X-Ray images using deep CNN and extreme learning machines stabilized by chimp optimization algorithm; Evolving deep convolutional neutral network by hybrid sine–cosine and extreme learning machine for real-time COVID19 diagnosis from X-ray images; Evolving deep learning convolutional neural networks for early COVID-19 detection in chest X-ray images; Pulmonary Diffuse Airspace Opacities Diagnosis from Chest X-Ray Images Using Deep Convolutional Neural Networks Fine-Tuned by Whale Optimizer.
  • Afterward, they have to answer the question: why do they use proposed model instead of the aforementioned algorithms.
  • Why is the proposed model the best solution to the problem posed?
  • How can the proposed solution be compared with other possible solutions such as those mentioned above, among others?
  • In the manuscript, there is no trace of the experimental setup and parameter settings for the algorithms. In fact, there is no sensitivity analysis for the proposed algorithm (can be found in: Real-Time COVID-19 Diagnosis from X-Ray Images Using Deep CNN and Extreme Learning Machines Stabilized by Chimp Optimization Algorithm; Evolving Deep Convolutional Neural Network by Hybrid Sine-Cosine and Extreme Learning Machine for Real-time COVID19 Diagnosis from X-Ray Images). These are very important aspects in any experiment with parameters.
  • The discussion of the results needs to include the strengths and weaknesses of the proposed algorithm.
  • It is worthy of mentioning that this method can be enhanced in figures using optimization techniques such as using the recent optimizer:  Chimp optimization algorithm, expert system with application.

Reviewer 2 Report

Many thanks for the opportunity to review the Manuscript entitled: Deep Transfer Learning on Multi-Label Classification of Chest 2 X-ray Images. The manuscript describes the effects of a variety of transfer learning approaches on deep CNN models for multi-label classification of chest x-ray images

I suggest these corrections:

  • Please edit the typo error chest x-ray into chest X-ray in your manuscript using always the same style
  • In line 117 of the Target data set specify the 7 categories
  • The table 2 should be improved avoiding some redundant features. Therefore: 

In the table 2 authors report 4 subcategories for aortic atherosclerosis/calcification that are very similar however I suggest to consider only aortic arc atherosclerosis plaque

 In the arterial curvature author considered aortic curvature and thoracic vertebral curvature. However, they are not radiological features. Therefore, in the arterial curvature should be include other radiological features as thoracic aortic ectasia

Tuberculosis is not a radiological diagnosis but some radiological features are suggestive of tuberculosis as: consolidations in the upper lobes or lung cavities. Therefore, these features should be considered for tuberculosis

Please divide the conclusion from the discussion